# Deep Neural Networks and the Tree of Life

**Yan Wang,**[*] **Kun He**[†]
Computer Science Department
Huazhong University of Science and Technology
{yanwang, brooklet60}@hust.edu.cn

**John E. Hopcroft, Yu Sun**
Computer Science Department
Cornell University
{jeh, ys646}@cs.cornell.edu

## Abstract

In Evolutionary Biology, species close in the tree of evolution are identified by similar visual features. In computer vision, deep neural networks perform image classification by learning to identify similar visual features. This leads to an interesting question: is it possible to leverage the advantage of deep networks to construct a tree of life? In this paper, we make the first attempt at building the phylogenetic tree diagram by leveraging the high-level features learned by deep neural networks. Our method is based on the intuition that if two species share similar features, then their cross activations in the softmax layer should be high. Based on the deep representation of convolutional neural networks trained for image classification, we build a tree of life for species in the image categories of ImageNet. Further, for species not in the ImageNet categories that are visually similar to some category, the cosine similarity of their activation vectors in the same layer should be high. By applying the inner product similarity of the activation vectors at the last fully connected layer for different species, we can roughly build their tree of life. Our work provides a new perspective to the deep representation and sheds light on possible novel applications of deep representation to other areas like Bioinformatics.

## 1 Introduction

Deep learning transforms the data into compact intermediate representations akin to principal components, and derives layered structures by removing the redundancy in representations (Li Deng, 2014). In recent years, deep learning has demonstrated great success with significant improvement in various artificial intelligence applications, including speech recognition (Sak et al., 2015), image recognition (Ciresan et al., 2012; Cir; Krizhevsky et al., 2012), and natural language processing (Vinyals et al., 2015; Socher et al., 2013).

Convolutional Neural Networks (CNNs) are mainly designed for image and video recognition. Typical CNN architecture alternates convolutional layers and pooling layers, followed by several fully connected or sparsely connected layers with a final softmax as the classification layer. Milestones include the 16-layer AlexNet (Krizhevsky et al., 2012), the 19-layer VGG (Simonyan & Zisserman, 2014), and the 22-layer GoogleNet (Szegedy et al., 2015). By adding identity function as a short cut, He et al. (2015) are able to build a substantially deeper ResNet with 152 layers, which received the first place on the ILSVRC 2015 image classification task (Russakovsky et al., 2015). Other very deep networks include the highway network with depths up to 100 layers (Srivastava et al., 2015). Eldan & Shamir (2016) provide a theoretical justification that reveals the utility of having deeper networks rather than wider networks, implying that future progress will lead to the development of even deeper networks.

**Understanding the deep representations of neural networks** has become increasingly difficult as the state-of-the-art models have more layers. This problem is important because it will help us understand the intrinsic mechanism of deep neural networks and explore possible novel applications based on the understanding. Ballester & de Araújo (2016) show how CNNs, trained to identify objects primarily in photos, could be used for abstract sketch recognition. Gatys et al. (2015a;b) utilize

---

[*]The first three authors contribute equally.
[†]Corresponding author.

the correlations between feature maps to synthesize natural textures and transfer artistic style with high perceptual quality. In Bioinformatics, deep neural networks are used for the analysis of medical images for cancer detection (Cirean et al., 2013) as well as drug discovery and toxicology (Dahl et al., 2014; Ramsundar et al., 2015; Wallach et al., 2015). A deep-learning approach based on the autoencoder architecture has been adopted to predict Gene Ontology annotations and gene-function relationships (Chicco et al., 2014).

**The Tree of Life**   refers to the compilation of a comprehensive phylogenetic (or evolutionary) database rooted at the last universal common ancestor of life on Earth. Over the course of hundreds of millions of years, the splitting and subsequent divergence of lineages has produced the tree of life, which has as its leaves the many species of organisms (Darwin, 1859). Here we refer to a phylogenetic tree, evolutionary tree or tree of life as a branching diagram showing the inferred genealogical relationships (Evaluate how close two species are in the evolutionary history, as evaluated by observed heritable traits, such as DNA sequences) among various biological species (Hug et al., 2016). This is an important problem in evolutionary biology and many attempts have been made (Darwin, 1859; Doolittle & Bapteste, 2007; Bapteste et al., 2009; Edwards, 2009). Originally tree of life was manually built based on the understanding of the evolution history or the visual similarity of the species. Today modern techniques have been applied based on the gene similarity.

**Our contributions**   are two-fold:

1) Provides a potential solution to the important problem of constructing a biology evolutionary tree.

We propose a novel approach in constructing a tree of life using the deep representation of CNNs trained for image classification. We conjuncture that the hierarchical feature representation learned by deep networks can be leveraged to quantify the visual similarity of the species. In this way, we might be able to construct tree of life using their feature similarity.

2) Gives insight into the representations produced by deep neural networks.

We conjecture that if images of two training categories share some similar features, then their cross activations in the softmax layer should be high. Hence we could evaluate the genetic distance of species within the training categories. Based on the deep representation of several typical CNNs, AlexNet (Krizhevsky et al., 2012), VGG (Simonyan & Zisserman, 2014) and ResNet (He et al., 2015) that are trained for ImageNet classification, we construct tree of life for dozens of species in the thousands of ImageNet categories of the training dataset.

For species not in the training categories that are visually similar to some species in the training dataset, could we still utilize their deep representation in order to judge the relationship among different species? We conjuncture that they show high cosine similarity of the activation vectors in high-level layers. By applying the inner product similarity of the activation vectors at the last fully connected layer for different species, we present empirical evidence that through transfer learning we could roughly construct their tree of life.

Experiments show that the proposed method using deep representation is very competitive to human beings in building the tree of life based on the visual similarity of the species. We also try on networks at different epochs during the training, and the quality of tree of life increases over the course of training. The performance among the three networks, AlexNet, VGG and ResNet, improves with the improvement of their classification quality.

## 2   THE PROPOSED METHOD

### 2.1   DATA COLLECTION

We have two important criterions in mind while constructing our image dataset. 1) We would like each image category, which corresponds to a node in the tree (i.e. a species), to have enough samples such that a statistic from the network activations is reasonably robust to noise. 2) There exists a ground truth hierarchy on the image categories, so we can objectively evaluate the effectiveness of our method.

Fortunately, the ImageNet 2012 Classification dataset provides the raw material we need. This dataset contains 1000 categories of common life objects, and each category contains 1000 images as the training data. Also, those categories correspond exactly to nodes in the WordNet hierarchy. WordNet (Miller, 1995) is a large lexical database of English, where words are grouped into sets of cognitive synonyms (synsets), each expressing a distinct concept and synsets are interlinked by means of conceptual-semantic and lexical relations.

For the reference network, we select three popular CNNs (AlexNet, VGG-16 and ResNet-152) trained on ImageNet. The top 5 classification errors of AlexNet, VGG and ResNet are $15.3\%$, $9.9\%$ and $6.7\%$ respectively. So they all learn the features of the images very well and we could leverage their deep representations for the ToL construction.

To find a small branch of the phylogenetic tree in order to do the reconstruction, we choose a set $\mathcal{A}$ of genealogically close species (species close in the evolutionary tree of life as evaluated by the branch distance) from the 1000 ImageNet categories. And for each category $A \in \mathcal{A}$, we use all the 1000 images from the training dataset to get robust result.

For the ground truth, in the smallest WordNet subtree that contains $\mathcal{A}$: 1) we could just consider the categories in $\mathcal{A}$ and their positions in this WordNet subtree and build a smallest ground truth tree $T_{\mathcal{A}}^1$. 2) we could additional consider some categories outside $\mathcal{A}$ in this WordNet subtree. Then the ground truth tree $T_{\mathcal{A}}^2$ contains some categories outside the ImageNet training categories. Note that nodes in $T_{\mathcal{A}}^1$ is basically the intersection of nodes in $T_{\mathcal{A}}^2$ and nodes in the 1000 ImageNet categories. For each category outside the 1000 training categories, we also use the 1000 images from the ImageNet database [1].

## 2.2 SIMILARITY EVALUATION

We input all selected images for species in $T_{\mathcal{A}}^1$ or $T_{\mathcal{A}}^2$ to a reference network and execute the feed forward pass. The feature maps (i.e. the activation vectors) of the last fully connected (FC) layer and the softmax layer are used to build the distance matrix.

1) **The Probability Method.** For $T_{\mathcal{A}}^1$, each class is in the training set and their ground truth labels are among the ones represented by the softmax layer. So we utilize the probability distribution of the images at the softmax layer in order to build a distance matrix. Specifically, for two classes of images $A$ and $B$ in the categories of $\mathcal{A}$, we consider their cross activations in the softmax layer. For each image $a \in A$, we obtain the predicted probability $P_{a2B}$ that this image belongs to node $B$, and we calculate the average of these values, named $P_{A2B}$.

$$P_{A2B} = \sum_{a \in A} P_{a2B} \tag{1}$$

For each image $b \in B$, we obtain the predicted probability $P_{b2A}$ that this image belongs to node $A$, and we calculate the average of these values, named $P_{B2A}$.

$$P_{B2A} = \sum_{b \in B} P_{b2A} \tag{2}$$

The closer the genealogical relationship of $A$ and $B$, the higher the cross predicted probability value should be. As the cross confidence is close to zero, we use the logistic function to enlarge the value. Then we add "$-$" to assign lower value to closer species and to keep the value nonnegative.

$$D_{AB} = \begin{cases} 0 & \text{if } A = B \\ -log(0.5P_{A2B} + 0.5P_{B2A}) & \text{if } A \neq B \end{cases} \tag{3}$$

2) **The Inner Product Method.** For $T_{\mathcal{A}}^2$, as some species are not in the 1000 classification categories, we use the centroid vector of the activations at the last fully connected (FC) layer for each species, and calculate the dot product of the two unitized centroid vectors to get their cosine similarity. Then we add "$-$" to assign lower value to closer species.

$$D_{AB} = -log \left( \frac{v_A \cdot v_B}{||v_A|| \, ||v_B||} \right) \tag{4}$$

---

[1]The only exception is for Bassarisk which only contains 694 images.

## 2.3 Constructing the Tree of Life

Based on the distance matrix, we have three methods, namely "Approximation Central Point", "Minimum Spanning Tree", and "Multidimensional Scaling", to construct a tree of life.

1) **The "Approximation Central Point"(ACP) based method.** In the ACP based method, we build a tree bottom up by recursively merging two species points, say $A$ and $B$, with the smallest distance, and setting the distance of the new point to other points as the average distance of $A$ and $B$ to other points respectively.

2) **The "Minimum Spanning Tree" (MST) based method.** In the MST based method, we first construct a Minimum Spanning Tree (MST) based on the distance matrix. Then we build a tree from the root to the leaves, recursively split the current MST subtree into two parts by removing its longest edge until there is only one node in each subtree. In this way we build a "tree" with all the leaves corresponding to the species and closest species are splitted in the end.

3) **The "Multidimensional Scaling"(MDS) based method.** In the MDS based method, according to $D$, we know distances among the points which corresponds to the species. We first apply the MDS (Multi-Dimensional Scaling) (Borg & Groenen, 2005) algorithm to do dimension reduction and project the species points into a two dimensional subspace. Then we build a tree bottom up by recursively merging two points with the smallest Euclidean distance in the two dimensional subspace and regard the midpoint of the two merging points as the new representative point.

Our following experiments show that MST and MDS show similar performance but ACP is considerably weaker.

## 3 Experiments and Analysis

We conduct a plenty set of experiments to build several branches of the phylogenetic trees of different granularity. To test whether our method could distinguish tiny visual differences, we first choose genealogically very close species, such as a set of fish species or a set of canine species, and construct their tree of life. Then, to test whether our method has good scalability for larger species, such as dog, cat, fish, etc., we choose 39 different large species to build a more general tree of life and verify whether different breeds of one large species like dogs could be grouped together. In addition, to evaluate the ability of constructing hierarchical trees based on the visual similarity of images outside the Biology, we choose some vehicle categories from the ImageNet dataset (Russakovsky et al., 2015) and build a vehicle tree.

For the methods, we use the probability method in Section 2.2 to build the distance matrix, and apply ACP, MST, and MDS based methods to build the tree of life. For the inner product method in Section 2.2, the results is slightly weaker, but it can deal with species or categories outside the training set. For details of inner product method, the readers are referred to the Appendix.

### 3.1 Constructing Fine-grained Tree of Life

To construct fine-grained tree of life, we select several fish species of high visual similarity and test whether we could identify the tiny differences of the features. We pick six fish species from the ImageNet training set and for each species, we input all the 1000 images in the training dataset to the ResNet network.

Figure 1 shows that the tree of life constructed by MST and MDS coincides with the hierarchial tree built on WordNet. The hierarchical tree constructed by ACP does not coincide with the ground truth at all. The reason may be that in any triangle $ABC$, the edge length from $A$ to the median of $BC$, say $D$, is shorter than the average length of edge $AB$ and $AC$. If $A$ is far more from symmetric as evaluated by edge $BC$, the recalculated distance of $AD$ does not accurately represent the distance of $A$ to the merged set of $\{B, C\}$.

Our results demonstrate that deep CNNs could capture the local features as well as the global features simultaneously. As to rebuild tree of life for genealogically close species, we need both features of different granularity like the animal's size, skin texture and shape. For instance, the texture of a

lionfish is very similar to that of a goldfish, then we need other features like the size to distinguish the two species.

As another example, we choose 11 very similar canine species and build a relatively lager tree, as illustrated in Figure 3. We can correctly build the canine tree, possibly according to their fur texture and shape features. The reconstructed quality is as good as what human beings could reconstruct based on the visual similarity.

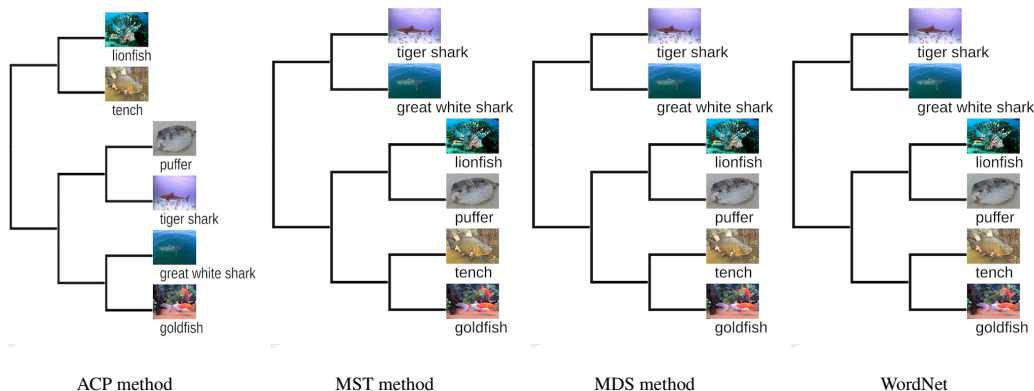

Figure 1: Trees of life for fish species. The first three trees are constructed by our methods, and the fourth tree is the ground truth using WordNet. The hierarchy of MST and MDS coincides with that of the WordNet.

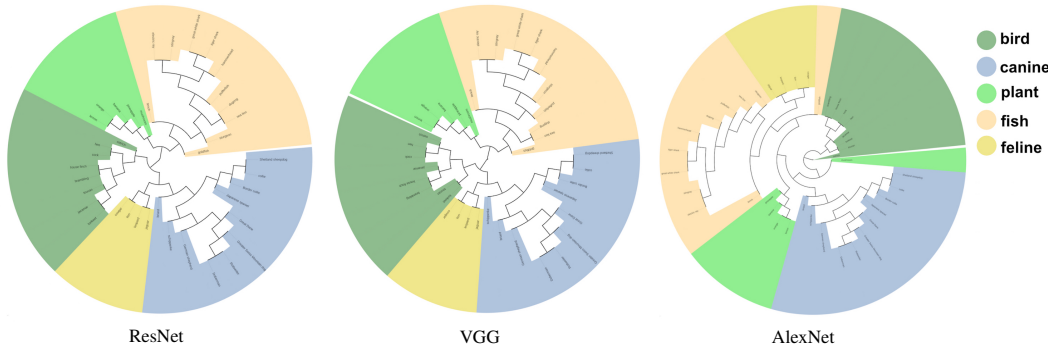

Figure 2: Constructed tree of life for families of species by different networks. Species of the five families are in different colors. ResNet and VGG can correctly cluster the species but AlexNet does not. Build by MST based method.

## 3.2 CONSTRUCTING COARSE-GRAINED TREE OF LIFE

Figure 2 shows the coarse-grained tree of life for clustering species of different families by different networks: ResNet, VGG and AlexNet. We pick 38 species from five families: bird, canine, plant, fish and feline.ResNet and VGG can correctly cluster the species by families, while AlexNet has makes some mistakes. This result indicates that deep networks with higher classification quality learn the deep representations better, such that the Tree of Life built based on the deep representation also have different reconstruction quality.

To show that we not only correctly cluster the species, but also ensure the correct hierarchy within each family, we further construct a tree containing 20 species of five families, as illustrated in Figure 4.

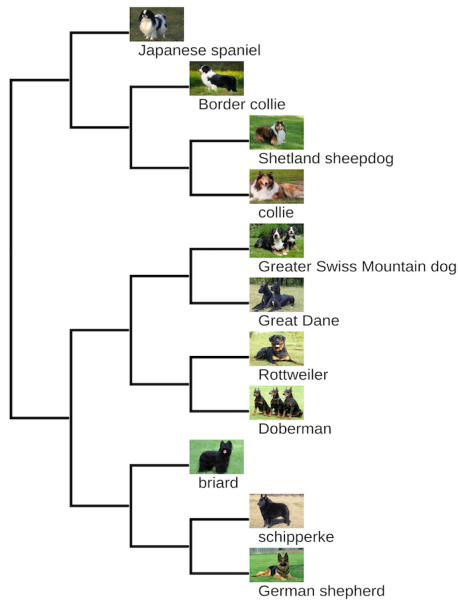

Figure 3: A constructed tree of life for 11 canine species. Closer species show shorter distance. Build by MDS based method.

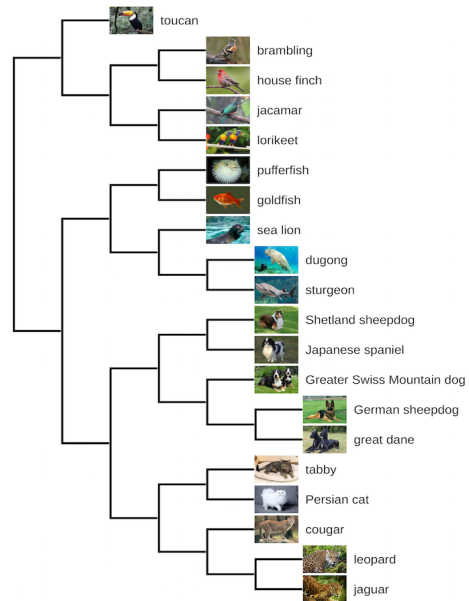

Figure 4: A constructed small tree of life for different families of species. We not only correctly cluster each family of species, but also present correct hierarchy of the species within each family. Build by MDS based method.

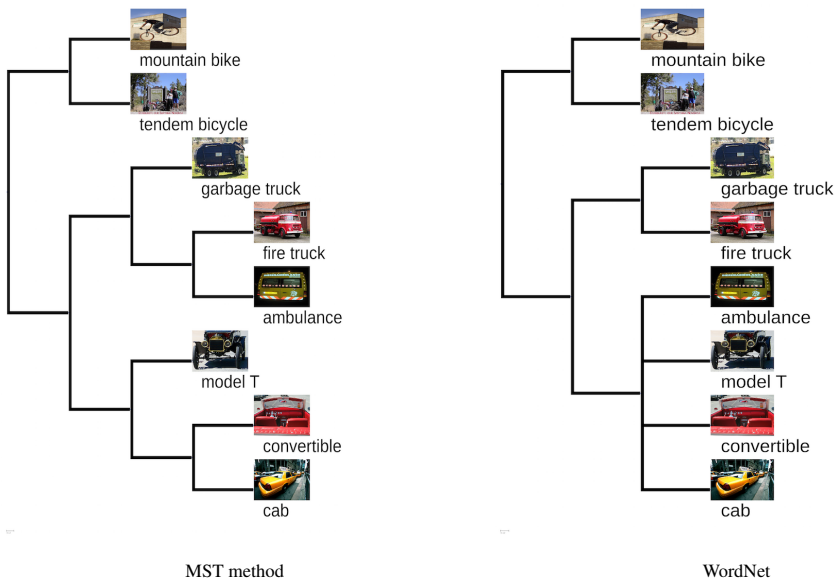

MST method                                        WordNet

Figure 5: A constructed vehicle tree. Our result looks more reasonable than that of the WordNet. Build by the MDS method.

## 3.3 Constructing a vehicle tree

To show the ability of building hierarchical tree for other objects other than animals, we pick eight vehicle categories from the ImageNet training set. Vehicles are very different from the animals.

Their shapes are kind of fixed and they can only do certain motions like going forward or turning around. Images of vehicles do not embed abundant features as the animal images do.

Nevertheless, our method still output good results, as shown in Figure 5. We cluster the ambulance, fire truck and garbage truck together, all of which have big carriages. While in WordNet, the ambulance is close to model T, convertible and cab, but the three do not have carriage and they are much smaller than ambulance. Our result is more reasonable than the WordNet provides.

## 4 CONCLUSION

By leveraging the similarity of features extracted automatically by deep learning techniques, we build a tree of life for various biological species, either belonging to the training categories or not. The results are highly competitive to the level of human beings in building the tree of life based on the visual similarity of the images. Our work provides new understandings to the deep representation of neural networks and sheds light on possible novel applications of deep learning in the area of Bioinformatics. An intriguing future work would be on how to utilize deep learning techniques to build a more delicate tree of life based on the gene similarity of the species.

## ACKNOWLEDGMENTS

This research work was supported by US Army Research Office(W911NF-14-1-0477) and National Science Foundation of China(61472147).

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

## APPENDIX

To test the inner product method in Section 2.2, that can build tree of the species not in the training set, we select 5 species not in the training set and 14 species in the training set. We choose 1000 images for each species except for Bassarisk which only contains 694 images. We show the results on ResNet using the MDS based method. Figure 6 illustrates the result.

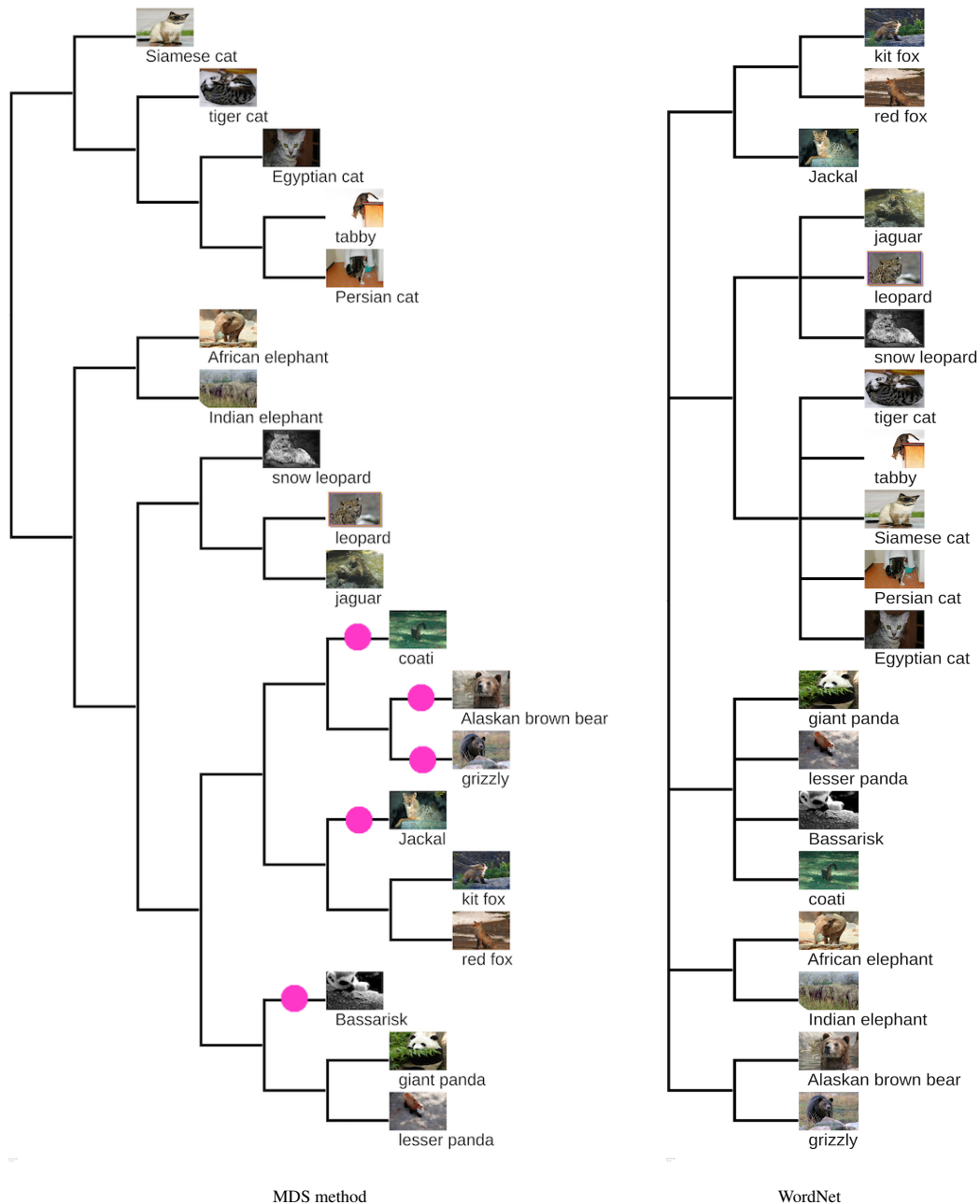

MDS method                                  WordNet

Figure 6: Constructing tree of life containing some species not in training set (marked by pink point). We use inner product method to build the distance matrix. Only coati is in the wrong leaf of the tree.

