# Peer review of "Deep Neural Networks and the Tree of Life"

_ICLR 2017 — rejected_

[Official Review · AnonReviewer1 · rating 4 · confidence 4 · 17 Dec 2016]
**Intellectually interesting but I'm not sure what the real contribution is**

I like this paper in that it is a creative application of computer vision to Biology. Or, at least, that would be a good narrative but I'm not confident biologists would actually care about the "Tree of Life" built from this method. There's not really any biology in this paper, either in methodology or evaluation. It boils down to a hierarchical clustering of visual categories with ground truth assumed to be the WordNet hierarchy (which may or may not be the biological ground truth inheritance relationships between species, if that is even possible to define -- it probably isn't for dog species which interbreed and it definitely isn't for vehicles) or the actual biological inheritance tree or what humans would do in the same task. If we're just worried about visual relationships and not inheritance relationships then a graph is the right structure, not a tree. A tree is needlessly lossy and imposes weird relationships (e.g. ImageNet has a photo of a "toy rabbit" and by tree distance it is maximally distant from "rabbit" because the toy is in the devices top level hierarchy and the real rabbit is in the animal branch. Are those two images really as semantically unrelated as is possible?). Our visual world is not a hierarchy. Our biological world can reasonably be defined as one. One could define the task of trying to recover the biological inheritance tree from visual inputs, although we know that would be tough to do because of situations like convergent evolution. Still, one could evaluate how well various visual features can recover the hierarchical relationship of biological organisms. This paper doesn't quite do that. And even if it did, it would still feel like a bit of a solution in search of a problem. The paper says that this type of exercise can help us understand deep features, but I'm not sure sure how much it reveals. I guess it's a fair question to ask if a particular feature produces meaningful class-to-class distances, but it's not clear that the biological tree of life or the wordnet hierarchy is the right ground truth for that (I'd argue it's not).

Finally, the paper mentions human baselines in a few places but I'm not really seeing it. "Experiments show that the proposed method using deep representation is very competitive to human beings in building the tree of life based on the visual similarity of the species." and then later "The reconstructed quality is as good as what human beings could reconstruct based on the visual similarity." That's the extent of the experiment? A qualitative result and the declaration that it's as good as humans could do?

[Official Review · AnonReviewer2 · rating 4 · confidence 4 · 19 Dec 2016]
**Nice application of using deep features but lack technical novelty**

This paper introduces a hierarchical clustering method using learned CNN features to build 'the tree of life'. The assumption is that the feature similarity indicates the distance in the tree. The authors tried three different ways to construct the tree: 1) approximation central point 2) minimum spanning tree and 3) multidimensional scaling based method. Out of them, MDS works the best. It is a nice application of using deep features. However, I lean toward rejecting the paper because the following reasons:

1) All experiments are conducted in very small scale. The experiments include 6 fish species, 11 canine species, 8 vehicle classes. There are no quantitative results, only by visualizing the generated tree versus the wordNet tree. Moreover, the assumption of using wordNet is not quite valid. WordNet is not designed for biology purpose and it might not reflect the true evolutionary relationship between species. 
2) Limited technical novelty. Most parts of the pipeline are standard, e.g. use pretrained model for feature extraction, use previous methods to construct hierarchical clustering. I think the technical contribution of this paper is very limited.

[Public Comment · (anonymous) · 19 Dec 2016]
**Is this something useful for biologist ? or for computer vision researcher ?**

I think the idea is interesting, but is that something useful to construct the tree based on visual features? 

Moreover, in the paper, you mentioned that, it is based on the conjecture that,
"if images of two training categories share some similar features, then their cross activations in the softmax layer should be high".

To me , this is something like the Neural style transfer,

[Official Review · AnonReviewer3 · rating 3 · confidence 5 · 20 Dec 2016]
**concerns about both contributions**

The paper presents a simple method for constructing a visual hierarchy of ImageNet classes based on a CNN trained on discriminate between the classes. It investigates two metrics for measuring inter-class similarity: (1) softmax probability outputs, i.e., the class confusion matrix, and (2) L2 distance between fc7 features, along with three methods for constructing the hierarchy given the distance matrix: (1) approximation central point, (2) minimal spanning tree, and (3) multidimensional scaling of Borg&Groenen 2005.

There are two claimed contributions: (1) Constructs a biology evolutionary tree, and (2) Gives insight into the representations produced by deep networks. 

Regarding (1), while the motivation of the work is grounded in biology, in practice the method is based only on visual similarity. The constructed trees thus can’t be expected to reflect the evolutionary hierarchy, and in fact there are no quantitative experiments that demonstrate that they do. 

Regarding (2), the technical depth of the exploration is not sufficient for ICLR. I’m not sure what we can conclude from the paper beyond the fact that CNNs are able to group categories together based on visual similarities, and deeper networks are able to do this better than more shallow networks (Fig 2).

In summary, this paper is unfortunately not ready for publication at this time.

[Final Decision · Program Chairs · 06 Feb 2017]
**ICLR committee final decision**

The reviewers agree that the paper provides a creative idea of using Computer Vision in Biology by building "the tree of life". However, they also agree that the paper in its current form is not ready for publication due to limited novelty and unclear impact/application. The authors did not post a rebuttal to address the concerns. The AC agrees with the reviewers', and encourages the authors to improve their manuscript as per reviewers' suggestions, and submit to a future conference.